# Combined association of obesity and other cardiometabolic diseases with severe COVID-19 outcomes: a nationwide cross-sectional study of 21 773 Brazilian adult and elderly inpatients

Natanael de Jesus Silva ![ORCID] ,[1,2] Rita de Cássia Ribeiro-Silva,[1,2,3] Andrêa Jacqueline Fortes Ferreira,[1,2,4] Camila Silveira Silva Teixeira,[1,2,4] Aline Santos Rocha,[1,2,3] Flávia Jôse Oliveira Alves,[1,2,4] Ila Rocha Falcão,[2] Elizabete de Jesus Pinto,[2,5] Carlos Antônio de Souza Teles Santos,[1,2,6] Rosemeire Leovigildo Fiaccone,[1,2,7] Maria Yury Travassos Ichihara,[1,2] Enny S Paixão ![ORCID] ,[1,2,8] Mauricio L Barreto[1,2,4]

For numbered affiliations see end of article.

**Correspondence to**
Natanael de Jesus Silva;
silva_natanael@hotmail.com

## ABSTRACT

**Objectives** To investigate the combined association of obesity, diabetes mellitus (DM) and cardiovascular disease (CVD) with severe COVID-19 outcomes in adult and elderly inpatients.

**Design** Cross-sectional study based on registry data from Brazil's influenza surveillance system.

**Setting** Public and private hospitals across Brazil.

**Participants** Eligible population included 21 942 inpatients aged ≥20 years with positive reverse transcription-PCR test for SARS-CoV-2 until 9 June 2020.

**Main outcome measures** Severe COVID-19 outcomes were non-invasive and invasive mechanical ventilation use, intensive care unit (ICU) admission and death. Multivariate analyses were conducted separately for adults (20–59 years) and elders (≥60 years) to test the combined association of obesity (without and with DM and/or CVD) and degrees of obesity with each outcome.

**Results** A sample of 8848 adults and 12 925 elders were included. Among adults, obesity with DM and/or CVD showed higher prevalence of invasive (prevalence ratio 3.76, 95% CI 2.82 to 5.01) and non-invasive mechanical ventilation use (2.06, 1.58 to 2.69), ICU admission (1.60, 1.40 to 1.83) and death (1.79, 1.45 to 2.21) compared with the group without obesity, DM and CVD. In elders, obesity alone (without DM and CVD) had the highest prevalence of ICU admission (1.40, 1.07 to 1.82) and death (1.67, 1.00 to 2.80). In both age groups, obesity alone and combined with DM and/or CVD showed higher prevalence in all outcomes than DM and/or CVD. A dose–response association was observed between obesity and death in adults: class I 1.32 (1.05 to 1.66), class II 1.41 (1.06 to 1.87) and class III 1.77 (1.35 to 2.33).

**Conclusions** The combined association of obesity, diabetes and/or CVD with severe COVID-19 outcomes may be stronger in adults than in elders. Obesity alone and combined with DM and/or CVD had more impact on the risk of COVID-19 severity

### Strengths and limitations of this study

► This is the first study that describes the independent and combined relationship of obesity with COVID-19 severity in Brazil, one of the biggest epicentres of the pandemic worldwide.

► The study was based on registry data of a large nationwide sample of patients admitted, due to severe SARS-CoV-2 infection, to public and private hospitals across the country.

► The large sample size and data availability allowed us to analyse the combined association of obesity, diabetes and cardiovascular disease with severe COVID-19 outcomes, separately by age groups and controlled by important confounding variables, for example, underlying comorbidities.

► The cross-sectional study design does not allow causal inference, and generalisation of results must be taken with caution since only hospitalised cases of severe COVID-19 were included.

► As the study used routinely collected data, which have not been designed primarily for research purposes, they may bring well-known limitations related to missing, underestimation and potential misclassification.

than DM and/or CVD in both age groups. The study also supports an independent relationship of obesity with severe outcomes, including a dose–response association between degrees of obesity and death in adults.

## INTRODUCTION

The COVID-19 pandemic, caused by the SARS-CoV-2, as of 11 July 2021, has already

reached more than 185 million infected people and more than 4 million deaths in all continents.[1] Individuals with advanced age and chronic diseases, including cardiometabolic diseases, are considered groups at major risk of complications and severe illness from COVID-19.[2 3] Obesity has been shown as an independent risk factor for COVID-19 disease.[4–6] High body mass index (BMI) has been mentioned as a significant risk factor for COVID-19, according to early clinical reports from China,[7] Italy,[8] France,[9] Mexico[10] and the USA.[11] Several studies have demonstrated that obesity is leading to considerably worse COVID-19 outcomes, especially greater risk of hospital and intensive care unit (ICU) admission, invasive mechanical ventilation and death.[11–14]

The COVID-19 pandemic is rapidly spreading worldwide, especially in the Americas, where obesity is already a prevalent and important public health problem.[15 16] Brazil is currently one of the biggest epicentres of the COVID-19 pandemic worldwide, with more than 18.9 million cases and 528 000 deaths until 11 July 2021.[1] In 2018, the prevalence of adult overweight and obesity in Brazil was estimated at 55.7% and 19.8%, respectively.[17] This obesogenic profile of the Brazilian population contributes, among other factors, to the high prevalence of obesity-related diseases, such as type 2 diabetes mellitus (DM) and cardiovascular diseases (CVDs), in the country.[18] The fact that individuals with obesity also have more comorbidity diseases, which are either risk factors for COVID-19 severity and death, makes obesity particularly ominous in COVID-19 disease.[10–13]

Several characteristics that can influence the clinical evolution of individuals infected with COVID-19, such as obesity, have been independently documented.[5 6 19] However, evidence is yet unclear on the combined effect that obesity and obesity-related comorbidities play in COVID-19 severity, especially, in different age groups. We aimed in this study to investigate the combined association of obesity, diabetes, and CVD with mechanical ventilation use, ICU admission, and death in a large sample of adult and elderly patients hospitalised with COVID-19 in Brazil. We also explored the independent association between degrees of obesity and the mentioned outcomes.

## METHODS
### Study design and population
This is a cross-sectional study based on registry data from SIVEP-Gripe (Influenza Epidemiological Surveillance Information System (Sistema de Informação de Vigilância Epidemiológica da Gripe)), an influenza surveillance system of Brazil's Ministry of Health. The study used the publicly available dataset of SIVEP-Gripe, which includes de-identified data on cases of severe acute respiratory syndrome across public and private hospitals in Brazil.[20] These data were obtained through the Rede CoVida's integrated data platform that has been built with official, open and authorised data for the production of knowledge about the COVID-19 pandemic.

Our study population was composed of patients aged 20 years or older, hospitalised for severe acute respiratory syndrome, with positive reverse transcription (RT)-PCR test for SARS-CoV-2 and final diagnosis of COVID-19 until 9 June 2020. Only cases with complete data on demographic characteristics and comorbidities and plausible BMI values were included in the study.

### Exposure variable
Obesity was defined as BMI equal to or greater than 30 $kg/m^2$, according to the cut-off points proposed by the WHO[21] and the Pan American Health Organization[22] for adults and elders, respectively. BMI was calculated by health professionals in the hospital from directly measured or patient self-reported height and weight. Guidelines for the collection and analysis of anthropometric data in health services have been previously standardised by the Ministry of Health.[23] BMI values <12 or >70 $kg/m^2$ were considered implausible and excluded.[24] Because BMI in the SIVEP-Gripe is mandatory and complete information for patients diagnosed or self-reported with obesity only, the nutritional status of all patients was confirmed by a dichotomous variable (no/yes) on the existence of obesity available in the dataset. Likewise, information on the existence of diabetes and any chronic CVD was obtained from dichotomous questions (no/yes), which were answered based on patient or family's report or medical diagnosis.

We created a polytomous four-category variable to evaluate the separate and combined exposure of obesity, diabetes and CVD: none/reference (no existence of obesity, diabetes and CVD), obesity (only existence of obesity), obesity+DM and/or CVD (existence of obesity with diabetes and/or CVD), and DM and/or CVD (existence of diabetes and/or CVD). We also analysed obesity in adults according to the following degrees of severity based on WHO reference[21]: no obesity (<30 $kg/m^2$), obesity class I (≥30–34.9 $kg/m^2$), obesity class II (≥35–39.9 $kg/m^2$) and obesity class III (≥40 $kg/m^2$). Due to the unavailability of BMI cut-off points to classify the degree of obesity in elders, this analysis was only performed for adults.

### Outcome variables
The severe COVID-19 outcomes were mechanical ventilation use, ICU admission and death. Information on the use of mechanical ventilation by the patient was obtained and analysed as a polytomous three-category variable (no use/use of non-invasive ventilation/ use of invasive ventilation). ICU admission was obtained and analysed as a dichotomous variable (no/yes). Death was analysed as a dichotomous variable based on the patient's endpoint outcome (cure/death).

### Covariates
Demographic and comorbidity information was selected as descriptive and confounding variables.[2] Age in years was calculated from birth and notification dates. Sex was

obtained as a dichotomous variable (female/male). The pre-existence of each comorbidity was also obtained as a dichotomous variable (no/yes): chronic pulmonary disease, asthma, chronic kidney disease, chronic haematological disease, neurological disease, chronic liver disease and immunodeficiency/immunosuppression.

## Statistical analysis

All analyses were subdivided into adults (≥20 and <60 years) and elders (≥60 years). For descriptive analyses, absolute and relative frequencies were calculated for the demographic and comorbidity variables according to the main exposure variable. Multinomial logistic regression models were conducted to test the association of obesity (without and with diabetes and/or CVD) with non-invasive and invasive mechanical ventilation use. To test the association of this exposure variable with ICU admission and death, simple logistic regression models were performed. Same models were analysed considering the degree of obesity as the main exposure variable for adults. Given the high prevalence of the analysed outcome that could overestimate OR,[25] crude and adjusted estimates were interpreted based on the prevalence ratio (PR) and 95% CIs. These estimates were obtained from logistic models using delta method, function 'prLogistic-Delta', which is implemented in R and available in the package 'prLogistic'. Adjusted models included a set of confounding variables selected according to the current literature on obesity and severe COVID-9 risk factors[26 27]: sex, age (years), and the pre-existence of chronic pulmonary disease, asthma, kidney disease, haematological disease, neurological disease, liver disease and immunodeficiency/immunosuppression. The models that tested the degrees of obesity were also adjusted for DM and CVD. All analyses were performed using Stata V.15.1 (Stata Corporation, College Station, USA) and R V.3.6.1 (R Foundation for Statistical Computing, Austria).

## Sensitivity analysis

Due the lack of detailed information on the comorbidities (eg, duration, severity), serious comorbidities such as chronic pulmonary diseases and immunosuppression were tested as exclusion criteria instead of confounding variables in a sensitivity analysis. Same multivariate logistic models, using 'death' as outcome variable, were conducted separately for adults and elders, excluding the cases of pulmonary diseases and immunosuppression.

## Patient and public involvement

As the study exclusively used publicly available de-identified data, it was not possible to involve patients or the public in the design, or conduct, or reporting, or dissemination plans of our research.

## RESULTS

During the study period, 21942 individuals registered in the SIVEP-Gripe were ≥20 years old, hospitalised, tested positive for SARS-CoV-2, and had complete demographic and comorbidity information (figure 1). Of these, 169 (0.8%) were excluded due to implausible values of BMI. Of the 21773 individuals included in the study, 8848 (40.3%) were adults aged between 20 and 59 years, and 12925 (59.6%) were elders aged 60 years or older. Since some patients were still hospitalised on the study endpoint date, information for some outcomes was incomplete. The study samples included in the analysis of each outcome were 8075 adults and 11829 elders for mechanical ventilation, 8414 adults and 12222 for ICU admission, and 6565 adults and 9943 elders for death.

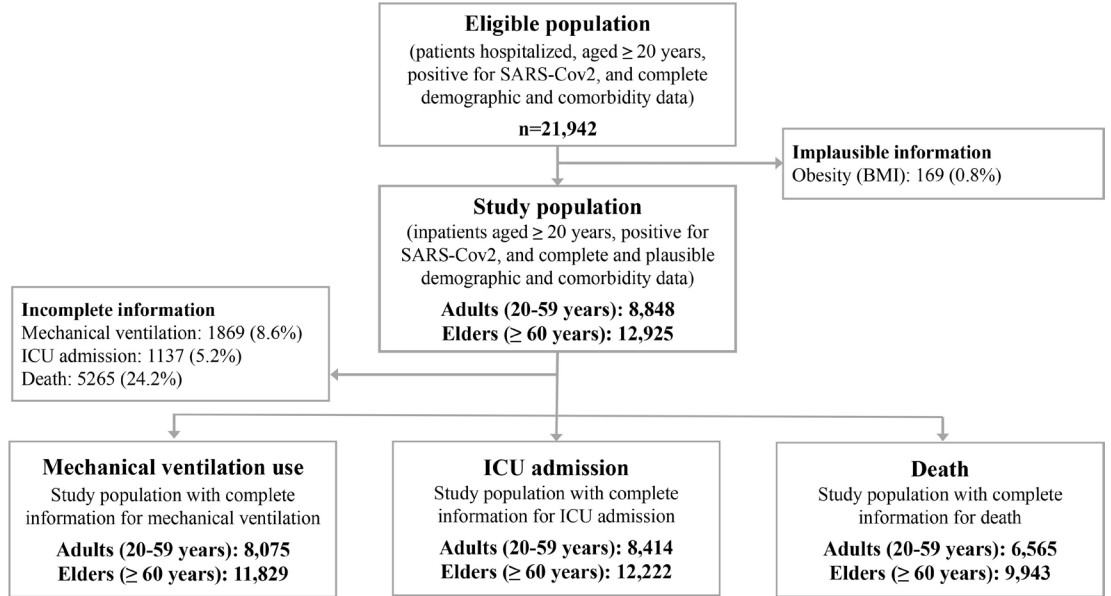

**Figure 1** Selection of the study population from SIVEP-Gripe. BMI, body mass index; ICU, intensive care unit; SIVEP-Gripe, Influenza Epidemiological Surveillance Information System.

Based on demographic and clinical characteristics, the analytical samples in each outcome were very similar to the overall study population and the excluded samples (online supplemental table 1).

The prevalence of obesity was 9.7% in adults and 3.5% in elders. The frequency of obesity without and with DM and/or CVD was, respectively, 4.6% and 5.1% in adults and 0.7% and 2.8% in elders. Non-invasive and invasive mechanical ventilation was, respectively, required by 45.0% and 21.2% of adults and 47.0% and 30.0% of elders. ICU admission was needed by 35.4% of adults and 43.6% of elders. Death occurred in 31.1% of adult and 63.0% of elderly patients (tables 1 and 2).

In the adjusted analyses for adults, obesity alone (without DM and CVD) was associated with an increased prevalence of invasive (PR 2.69, 95% CI 1.98 to 3.65) and non-invasive mechanical ventilation need (PR 2.13, 95% CI 1.64 to 2.78), ICU admission (PR 1.31, 95% CI 1.13 to 1.53), and death (PR 1.33, 95% CI 1.05 to 1.69) when compared with the group without obesity, DM, and CVD. Obesity with DM and/or CVD was associated with an even higher prevalence of invasive mechanical ventilation (PR 3.76, 95% CI 2.82 to 5.01) and non-invasive ventilation use (PR 2.06, 95% CI 1.58 to 2.69), ICU admission (PR 1.60, 95% CI 1.40 to 1.83) and death in adults (PR 1.79, 95% CI 1.45 to 2.21). The subgroup of adults with DM and/or CVD showed in general lower PRs for all analysed outcomes than the subgroups with the presence of obesity alone or combined (table 3).

Among elders, obesity without DM and CVD was independently associated with a higher prevalence of ICU admission (PR 1.40, 95% CI 1.07 to 1.82) and death (PR 1.67, 95% CI 1.00 to 2.80). To a lesser extent, obesity with DM and/or CVD was also associated with an increased prevalence of invasive mechanical ventilation need (PR 1.66, 95% CI 1.22 to 2.27), ICU admission (PR 1.37, 95% CI 1.19 to 1.59) and death (PR 1.39, 95% CI 1.07 to 1.80). Elders with DM and/or CVD had lower PRs for the analysed outcomes than the group of elders with obesity alone or combined (table 3).

In the analyses by the degree of obesity, we did not observe much difference in the prevalence of adverse outcomes, except for the prevalence of death that increased with the severity of obesity: class I 1.32 (95% CI 1.05 to 1.66), class II 1.41 (1.06 to 1.87) and class III 1.77 (1.35 to 2.33) (table 4).

The sensitivity analysis, excluding the cases of chronic pulmonary diseases and immunosuppression (online supplemental tables 2 and 3), showed no difference in the results when compared with the estimates described above which were instead adjusted for these comorbidities. Only small differences in the magnitude of the associations, but not in the direction and significance, were observed.

## DISCUSSION

This is the first study that describe the relationship of obesity and COVID-19 in Brazil, based on a large nationwide sample of adults and elders tested positive for SARS-CoV-2 and admitted to public and private hospitals. Our results highlight that obesity with DM and/or CVD was associated with higher rates of invasive mechanical ventilation use, ICU admission and death in adults, while obesity alone (without DM and CVD) was associated with higher rates of ICU admission and death among elders. In both age groups, obesity alone and obesity combined with DM and/or CVD had more impact on the risk of all severe COVID-19 outcomes than the subgroup with DM and/or CVD. The study also supports the independent association of obesity with the analysed outcomes and a dose–response association between degrees of obesity and death in adults.

Some mechanisms related to the role of obesity and related diseases in the worsening clinical condition of patients affected by SARS-CoV-2 have been pointed out: (1) greater body weight causes less elasticity of the chest wall and less total compliance of the respiratory system, leading to a restriction of the ventilation and the excursion of the diaphragm, making the airway management in patients with obesity difficult[28]; (2) obesity is associated with sleep apnoea syndrome and chronic obstructive pulmonary disease, which lead to surfactant dysfunction and impede the proper functioning of the airways[29]; (3) obesity is a metabolic and inflammatory disease, which is associated with the development or worsening of other chronic and endocrine comorbidities (eg, type 2 diabetes, hypertension, dyslipidaemia and CVD) that can modify innate and adaptive immune responses, making the immune system more vulnerable to infections and less responsive to antivirals and antimicrobial drugs[16]; (4) glycaemic decompensation, common in patients with obesity, is associated with impaired ventilation function.[29]

It is important to note that the COVID-19 pandemic imposes a double burden of disease, especially among the elderly individuals, since the prevalence of diabetes, hypertension, CVDs and other comorbidities associated with COVID-19 severity increases with age.[3 30] However, our study suggests that obesity combined with diabetes and/or CVD may offer higher risk of COVID-19 severity for adults, although the overall prevalence of diseases and rates of ICU admission and mortality were higher in elders. Obesity alone seemed to provide higher risk of severe outcomes, especially death, in elders. As 1.00 remains a plausible value for the PR, according to its CIs, we cannot conclude that the association of obesity alone with death in elders is clinically relevant. A larger sample would be needed to fully address this more.

Few studies to date have explored the combined and additional effect of obesity on COVID-19 severity.[13 31] A study investigated the patterns of multimorbidity among fatal cases of COVID-19 in Colombia.[31] Similar to our study, the authors found that obesity alone or with other diseases was associated with a higher risk of COVID-19 fatality among young people. Furthermore, a population-based study in Mexico observed that the addition of obesity to any number of comorbidities significantly

**Table 1** Demographic characteristics, comorbidities, hospitalisation outcomes and death according to the combined exposure of obesity (OB), diabetes mellitus (DM), and/or cardiovascular diseases (CVDs) in adults with severe COVID-19

| | Total | | None | | OB | | OB +DM and/or CVD | | DM and/or CVD | |
|---|---|---|---|---|---|---|---|---|---|---|
| | n | % | n | % | n | % | n | % | n | % |
| Overall | 8848 | 100.0 | 3161 | 35.7 | 409 | 4.6 | 452 | 5.1 | 4826 | 54.6 |
| Sex | | | | | | | | | | |
| Female | 3774 | 42.7 | 1511 | 40.0 | 165 | 4.4 | 199 | 5.3 | 1899 | 50.3 |
| Male | 5074 | 57.4 | 1650 | 32.5 | 244 | 4.8 | 253 | 5.0 | 2927 | 57.7 |
| Age | | | | | | | | | | |
| <40 years | 1976 | 22.3 | 1064 | 53.9 | 188 | 9.5 | 102 | 5.2 | 622 | 31.5 |
| ≥40 years | 6872 | 77.7 | 2097 | 30.5 | 221 | 3.2 | 350 | 5.1 | 4204 | 61.2 |
| Chronic pulmonary disease | | | | | | | | | | |
| No | 8502 | 96.1 | 2969 | 34.9 | 388 | 4.6 | 435 | 5.1 | 4710 | 55.4 |
| Yes | 346 | 3.9 | 192 | 55.5 | 21 | 6.1 | 17 | 4.9 | 116 | 33.5 |
| Asthma | | | | | | | | | | |
| No | 8184 | 92.5 | 2728 | 33.3 | 383 | 4.7 | 414 | 5.1 | 4659 | 56.9 |
| Yes | 664 | 7.5 | 433 | 65.2 | 26 | 3.9 | 38 | 5.7 | 167 | 25.2 |
| Chronic kidney disease | | | | | | | | | | |
| No | 8297 | 93.8 | 2958 | 35.7 | 399 | 4.8 | 434 | 5.2 | 4506 | 54.3 |
| Yes | 551 | 6.2 | 203 | 36.8 | 10 | 1.8 | 18 | 3.3 | 320 | 58.1 |
| Chronic haematological disease | | | | | | | | | | |
| No | 8710 | 98.4 | 3081 | 35.4 | 406 | 4.7 | 445 | 5.1 | 4778 | 54.9 |
| Yes | 138 | 1.6 | 80 | 58.0 | 3 | 2.2 | 7 | 5.1 | 48 | 34.8 |
| Chronic neurological disease | | | | | | | | | | |
| No | 8588 | 97.1 | 3014 | 35.1 | 406 | 4.7 | 442 | 5.2 | 4726 | 55.0 |
| Yes | 260 | 2.9 | 147 | 56.5 | 3 | 1.2 | 10 | 3.9 | 100 | 38.5 |
| Chronic liver disease | | | | | | | | | | |
| No | 8684 | 98.2 | 3083 | 35.5 | 406 | 4.7 | 443 | 5.1 | 4752 | 54.7 |
| Yes | 164 | 1.9 | 78 | 47.6 | 3 | 1.8 | 9 | 5.5 | 74 | 45.1 |
| Immunosuppression | | | | | | | | | | |
| No | 8276 | 93.5 | 2777 | 33.6 | 393 | 4.8 | 440 | 5.3 | 4666 | 56.4 |
| Yes | 572 | 6.5 | 384 | 67.1 | 16 | 2.8 | 12 | 2.1 | 160 | 28.0 |
| Mechanical ventilation* | | | | | | | | | | |
| No | 2727 | 33.8 | 1144 | 42.0 | 93 | 3.4 | 88 | 3.2 | 1402 | 51.4 |
| Non-invasive | 3634 | 45.0 | 1178 | 32.4 | 192 | 5.3 | 190 | 5.2 | 2074 | 57.1 |
| Invasive | 1714 | 21.2 | 529 | 30.9 | 101 | 5.9 | 150 | 8.8 | 934 | 54.5 |
| ICU admission* | | | | | | | | | | |
| No | 5438 | 64.6 | 2025 | 37.2 | 235 | 4.3 | 222 | 4.1 | 2956 | 54.4 |
| Yes | 2976 | 35.4 | 1007 | 33.8 | 163 | 5.5 | 216 | 7.3 | 1590 | 53.4 |
| Death* | | | | | | | | | | |
| No | 4525 | 68.9 | 1699 | 37.6 | 211 | 4.7 | 200 | 4.4 | 2415 | 53.4 |
| Yes | 2040 | 31.1 | 640 | 31.4 | 92 | 4.5 | 140 | 6.9 | 1168 | 57.3 |

OB (BMI ≥30 kg/m²).
*Mechanical ventilation (n=8075), ICU admission (n=8414) and death (n=6565).
BMI, body mass index; DM, diabetes mellitus; ICU, intensive care unit.

increased the risk of COVID-19 lethality.[13] Using a causally ordered mediation analysis, this study also found that 49.5% of the effect of diabetes on COVID-19 lethality was mediated by obesity, particularly in early-onset cases <40 years of age.

Other studies also suggest that obesity is independently associated with severe outcomes of COVID-19, regardless of age and other associated comorbidities.[11–14] A large study in Mexico[13] showed that patients with obesity had higher rates of ICU admission and were more likely to

**Table 2** Demographic characteristics, comorbidities, hospitalisation outcomes and death according to the combined exposure of obesity (OB), diabetes mellitus (DM), and/or cardiovascular diseases (CVDs) in elders with severe COVID-19

| | Total | | None | | OB | | OB +DM and/or CVD | | DM and/or CVD | |
|---|---|---|---|---|---|---|---|---|---|---|
| | n | % | n | % | n | % | n | % | n | % |
| Overall | 12 925 | 100.0 | 2837 | 21.9 | 91 | 0.7 | 358 | 2.8 | 9639 | 74.6 |
| Sex | | | | | | | | | | |
| Female | 5968 | 46.2 | 1232 | 20.6 | 52 | 0.9 | 209 | 3.5 | 4475 | 75.0 |
| Male | 6957 | 53.8 | 1605 | 23.1 | 39 | 0.6 | 149 | 2.1 | 5164 | 74.2 |
| Age | | | | | | | | | | |
| <80 years | 9355 | 72.4 | 2011 | 21.5 | 77 | 0.8 | 309 | 3.3 | 6958 | 74.4 |
| ≥80 years | 3570 | 27.6 | 826 | 23.1 | 14 | 0.4 | 49 | 1.4 | 2681 | 75.1 |
| Chronic pulmonary disease | | | | | | | | | | |
| No | 11 885 | 92.0 | 2494 | 21.0 | 85 | 0.7 | 325 | 2.7 | 8981 | 75.6 |
| Yes | 1040 | 8.1 | 343 | 33.0 | 6 | 0.6 | 33 | 3.2 | 658 | 63.3 |
| Asthma | | | | | | | | | | |
| No | 12 474 | 96.5 | 2687 | 21.5 | 90 | 0.7 | 336 | 2.7 | 9361 | 75.0 |
| Yes | 451 | 3.5 | 150 | 33.3 | 1 | 0.2 | 22 | 4.9 | 278 | 61.6 |
| Chronic kidney disease | | | | | | | | | | |
| No | 11 882 | 91.9 | 2608 | 22.0 | 85 | 0.7 | 311 | 2.6 | 8878 | 74.7 |
| Yes | 1043 | 8.1 | 229 | 22.0 | 6 | 0.6 | 47 | 4.5 | 761 | 73.0 |
| Chronic haematological disease | | | | | | | | | | |
| No | 12 728 | 98.5 | 2751 | 21.6 | 91 | 0.7 | 354 | 2.8 | 9532 | 74.9 |
| Yes | 197 | 1.5 | 86 | 43.7 | 0 | 0.0 | 4 | 2.0 | 107 | 54.3 |
| Chronic neurological disease | | | | | | | | | | |
| No | 11 871 | 91.9 | 2511 | 21.2 | 89 | 0.8 | 338 | 2.9 | 8933 | 75.3 |
| Yes | 1054 | 8.2 | 326 | 30.9 | 2 | 0.2 | 20 | 1.9 | 706 | 67.0 |
| Chronic liver disease | | | | | | | | | | |
| No | 12 734 | 98.5 | 2777 | 21.8 | 87 | 0.7 | 353 | 2.8 | 9517 | 74.7 |
| Yes | 191 | 1.5 | 60 | 31.4 | 4 | 2.1 | 5 | 2.6 | 122 | 63.9 |
| Immunosuppression | | | | | | | | | | |
| No | 12 303 | 95.2 | 2558 | 20.8 | 87 | 0.7 | 342 | 2.8 | 9316 | 75.7 |
| Yes | 622 | 4.8 | 279 | 44.9 | 4 | 0.6 | 16 | 2.6 | 323 | 51.9 |
| Mechanical ventilation* | | | | | | | | | | |
| No | 2725 | 23.0 | 626 | 23.0 | 18 | 0.7 | 70 | 2.6 | 2011 | 73.8 |
| Non-invasive | 5557 | 47.0 | 1164 | 21.0 | 38 | 0.7 | 141 | 2.5 | 4214 | 75.8 |
| Invasive | 3547 | 30.0 | 767 | 21.6 | 29 | 0.8 | 133 | 3.8 | 2618 | 73.8 |
| ICU admission* | | | | | | | | | | |
| No | 6898 | 56.4 | 1578 | 22.9 | 41 | 0.6 | 168 | 2.4 | 5111 | 74.1 |
| Yes | 5324 | 43.6 | 1107 | 20.8 | 44 | 0.8 | 181 | 3.4 | 3992 | 75.0 |
| Death* | | | | | | | | | | |
| No | 3684 | 37.1 | 823 | 22.3 | 21 | 0.6 | 95 | 2.6 | 2745 | 74.5 |
| Yes | 6259 | 63.0 | 1407 | 22.5 | 43 | 0.7 | 173 | 2.8 | 4636 | 74.1 |

OB (BMI ≥30 kg/m²).
*Mechanical ventilation (n=11 829), ICU admission (n=12 222) and death (n=9943).
BMI, body mass index; DM, diabetes mellitus; ICU, intensive care unit.

be intubated in relation to patients without obesity. This study also found a fivefold increased risk of mortality due to COVID-19 in patients with obesity.[13] In a hospital-based study in France, it was observed that BMI >35 kg/m² was associated with the need for invasive mechanical ventilation.[14]

Few studies to date have similarly found a dose–response association between degrees of obesity and

**Table 3** Combined association of obesity (OB), diabetes mellitus (DM), and/or cardiovascular disease (CVD) with non-invasive and invasive mechanical ventilation use, intensive care unit (ICU) admission, and death in adult and elderly patients hospitalised with severe COVID-19

| | Main exposure variable | Non-invasive mechanical ventilation* | | | | Invasive mechanical ventilation* | | | |
| | | Crude model | | Adjusted model† | | Crude model | | Adjusted model† | |
| | | PR | 95% CI | PR | 95% CI | PR | 95% CI | PR | 95% CI |
|---|---|---|---|---|---|---|---|---|---|
| Adults 20–59 years | None | 1.00 | | 1.00 | | 1.00 | | 1.00 | |
| | OB | 2.00 | 1.54 to 2.60 | 2.13 | 1.64 to 2.78 | 2.35 | 1.74 to 3.17 | 2.69 | 1.98 to 3.65 |
| | OB +DM and/or CVD | 2.10 | 1.61 to 2.73 | 2.06 | 1.58 to 2.69 | 3.69 | 2.78 to 4.89 | 3.76 | 2.82 to 5.01 |
| | DM and/or CVD | 1.44 | 1.29 to 1.60 | 1.35 | 1.20 to 1.51 | 1.44 | 1.26 to 1.64 | 1.32 | 1.14 to 1.52 |
| Elders ≥60 years | None | 1.00 | | 1.00 | | 1.00 | | 1.00 | |
| | OB | 1.14 | 0.64 to 2.01 | 1.22 | 0.69 to 2.16 | 1.31 | 0.72 to 2.39 | 1.43 | 0.78 to 2.61 |
| | OB +DM and/or CVD | 1.08 | 0.80 to 1.47 | 1.15 | 0.84 to 1.55 | 1.55 | 1.14 to 2.11 | 1.66 | 1.22 to 2.27 |
| | DM and/or CVD | 1.13 | 1.01 to 1.26 | 1.14 | 1.01 to 1.27 | 1.06 | 0.94 to 1.20 | 1.10 | 0.97 to 1.24 |
| | | ICU admission‡ | | | | Death§ | | | |
| | | Crude model | | Adjusted model† | | Crude model | | Adjusted model† | |
| | | PR | 95% CI | PR | 95% CI | PR | 95% CI | PR | 95% CI |
| Adults 20–59 years | None | 1.00 | | 1.00 | | 1.00 | | 1.00 | |
| | OB | 1.23 | 1.08 to 1.40 | 1.31 | 1.13 to 1.53 | 1.11 | 0.92 to 1.33 | 1.33 | 1.05 to 1.69 |
| | OB +DM and/or CVD | 1.48 | 1.33 to 1.65 | 1.60 | 1.40 to 1.83 | 1.50 | 1.30 to 1.74 | 1.79 | 1.45 to 2.21 |
| | DM and/or CVD | 1.05 | 0.99 to 1.12 | 1.03 | 0.95 to 1.12 | 1.19 | 1.10 to 1.29 | 1.16 | 1.03 to 1.30 |
| Elders ≥60 years | None | 1.00 | | 1.00 | | 1.00 | | 1.00 | |
| | OB | 1.26 | 1.02 to 1.55 | 1.40 | 1.07 to 1.82 | 1.06 | 0.89 to 1.27 | 1.67 | 1.00 to 2.80 |
| | OB +DM and/or CVD | 1.26 | 1.13 to 1.41 | 1.37 | 1.19 to 1.59 | 1.02 | 0.93 to 1.12 | 1.39 | 1.07 to 1.80 |
| | DM and/or CVD | 1.06 | 1.01 to 1.12 | 1.11 | 1.04 to 1.18 | 1.00 | 0.96 to 1.03 | 1.05 | 0.95 to 1.16 |

OB (BMI ≥30 kg/m$^2$).
*Crude and adjusted multinomial logistic regression models for mechanical ventilation use in adults (n=8075) and elders (n=11 829).
†Adjusted for sex, age in years, pulmonary disease, asthma, kidney disease, haematological disease, neurological disease, liver disease and immunosuppression.
‡Crude and adjusted logistic regression models for ICU admission in adults (n=8414) and elders (n=12 222).
§Crude and adjusted logistic regression models for death in adults (n=6565) and elders (n=9943).
BMI, body mass index; DM, diabetes mellitus; PR, prevalence ratio.

COVID-19 death.[32] Based on care records of 17 278 392 UK adults, the study showed that the risk of COVID-19 death increases independently with the degree of obesity: 30–34.9 kg/m$^2$ (HR 1.05), 35–39.9 kg/m$^2$ (1.40) and ≥40 kg/m$^2$ (2.66).[32] Other studies have evidenced the association of obesity with COVID-19 complications and death among adults.[12 33] A hospital-based study in New York City showed that morbid obesity (BMI ≥40 kg/m$^2$) is strongly and independently associated with death in hospitalised patients younger than 50 years.[33] Another study in New York City found a similar dose–response relationship between degrees of obesity and acute and critical care.[12] Patients less than 60 years old with BMI between 30 and 34.9 kg/m$^2$ (obesity class I) were 2.0 and 1.8 times more likely to be admitted for acute care (general hospital admission) and critical care (ICU admission or invasive ventilator), respectively, compared with individuals with BMI <30 kg/m$^2$. Patients of the same age group with BMI ≥35 kg/m$^2$ (obesity class II and III) showed 2.2 and 3.6 more chances of being hospitalised for acute and critical care, respectively.[12]

## Strengths and limitations

One of the greatest strengths of the study was the use of SIVEP-Gripe dataset. Because severe acute respiratory syndrome is a condition of compulsory notification in both public and private hospitals,[34] we have a nationwide representative sample of patients hospitalised for severe COVID-19 in Brazil. In addition, the large sample sizes allowed us to analyse adults and elders separately, as well as the degrees of obesity which dose–response association with death was evidenced. The availability of important confounding variables (sex, age and pre-existing comorbidities) to control the estimated associations, as well as hospital outcomes and mortality of COVID-19, was another differential of the study. Only patients with positive RT-PCR test for SARS-CoV-2 and final diagnosis for COVID-19 were included which gives greater precision on the studied population. The availability and use of data from health surveillance systems may be a lesson from Brazil that other countries can learn for obtaining routine and timely data to guide health systems and research in preparing and responding to pandemics before and during their course.

**Table 4** Independent association of degrees of obesity with non-invasive and invasive mechanical ventilation, intensive care unit (ICU) admission and death in hospitalised adults with severe COVID-19

| Main exposure variable | Non-invasive mechanical ventilation* | | | | Invasive mechanical ventilation* | | | |
|---|---|---|---|---|---|---|---|---|
| | Crude model | | Adjusted model† | | Crude model | | Adjusted model† | |
| | PR | 95% CI | PR | 95% CI | PR | 95% CI | PR | 95% CI |
| No obesity (<30 kg/m²) | 1.00 | | 1.00 | | 1.00 | | 1.00 | |
| Obesity class I (≥30–34.9 kg/m²) | 1.78 | 1.35 to 2.33 | 1.91 | 1.45 to 2.51 | 2.59 | 1.93 to 3.47 | 3.00 | 2.22 to 4.05 |
| Obesity class II (≥35–39.9 kg/m²) | 1.44 | 1.04 to 2.00 | 1.58 | 1.14 to 2.19 | 2.10 | 1.47 to 2.99 | 2.47 | 1.72 to 3.54 |
| Obesity class III (≥40 kg/m²) | 1.70 | 1.19 to 2.44 | 1.88 | 1.31 to 2.69 | 2.51 | 1.71 to 3.70 | 3.00 | 2.03 to 4.45 |
| | ICU admission‡ | | | | Death‡ | | | |
| | Crude model | | Adjusted model† | | Crude model | | Adjusted model† | |
| | PR | 95% CI | PR | 95% CI | PR | 95% CI | PR | 95% CI |
| No obesity (<30 kg/m²) | 1.00 | | 1.00 | | 1.00 | | 1.00 | |
| Obesity class I (≥30–34.9 kg/m²) | 1.31 | 1.17 to 1.47 | 1.42 | 1.23 to 1.64 | 1.11 | 0.94 to 1.31 | 1.32 | 1.05 to 1.66 |
| Obesity class II (≥35–39.9 kg/m²) | 1.34 | 1.16 to 1.54 | 1.46 | 1.23 to 1.74 | 1.16 | 0.95 to 1.42 | 1.41 | 1.06 to 1.87 |
| Obesity class III (≥40 kg/m²) | 1.32 | 1.14 to 1.54 | 1.45 | 1.20 to 1.74 | 1.33 | 1.10 to 1.59 | 1.77 | 1.35 to 2.33 |

Degrees of obesity defined by the WHO cut-off points.
*Crude and adjusted multinomial logistic regression models for mechanical ventilation use (n=8075).
†Adjusted for sex, age in years, diabetes mellitus, cardiovascular disease, pulmonary disease, asthma, kidney disease, haematological disease, neurological disease, liver disease and immunosuppression.
‡Crude and adjusted logistic regression models for ICU admission (n=8414) and mortality (n=6565).
PR, prevalence ratio.

The study also has some limitations that must be considered. Because this is a cross-sectional study, a causal association cannot be inferred. As we used routinely collected data, which have not been designed primarily for research purposes, they may bring well-known limitations related to missing, underestimation and potential misclassification. Obesity prevalence may have been underestimated due to the completeness of obesity and BMI data. Previous studies using SIVEP-Gripe data have also found a low prevalence of obesity in this population.[35 36] Better routine collection of height and weight data is still needed in clinical practice. Also, we believe that health professionals have adopted more the one method to collect weight and height information for BMI calculation, such as the patient's self-report and direct measure. Therefore, in addition to BMI which implausible values were checked and excluded, the classification of obesity was also confirmed from a dichotomous variable on the presence of obesity (no/yes). Although it is known that BMI does not distinguish between fat and lean body mass, and thus may lead to misclassification bias, BMI has been shown as a strong predictor of excess body fat and has been widely used in epidemiological studies.[15] Information for some outcomes was incomplete because some patients were still hospitalised on the study endpoint date. However, that did not represent a potential selection bias to our study. The analytical samples in each outcome had similar demographic and clinical characteristics to the overall study population and the excluded samples (online supplemental table 1). As information on smoking was not available and ethnicity/race was very incomplete in the SIVEP-Gripe dataset, they were not included in the analysis. Additional studies are needed to further explore the relationship between obesity and severe COVID-19, considering health risk behaviours and socioeconomic characteristics. Finally, the generalisation of results must be taken with caution since the study included only hospitalised cases of COVID-19.

## CONCLUSIONS

The combined association of obesity, diabetes, and/or cardiovascular disease with severe COVID-19 outcomes, especially ICU admission and death, may be stronger in adult than in elderly inpatients. In both age groups, obesity alone and obesity combined with DM and/or CVD had more impact on the risk of all severe COVID-19 outcomes than the subgroup with DM and/or CVD. The study also supports an independent relationship of obesity with the severe outcomes, including a dose–response association between degrees of obesity and death in adults. These findings suggest important implications for the clinical care of patients with obesity and severe COVID-19, such as the increased need of critical care and higher risk of death among these patients. Our study also supports the inclusion of people with obesity, independently of other pre-existing comorbidities and age, in the high-risk and vaccine-priority groups for protection from SARS-CoV-2 infection.

**Author affiliations**
[1]Rede CoVida, Salvador, BA, Brazil
[2]Centre for Data and Knowledge Integration for Health, Oswaldo Cruz Foundation, Salvador, BA, Brazil
[3]School of Nutrition, Federal University of Bahia, Salvador, BA, Brazil
[4]Institute of Collective Health, Federal University of Bahia, Salvador, BA, Brazil

[5]Center for Health Sciences, Federal University of Reconcavo da Bahia, Santo Antônio de Jesus, BA, Brazil

[6]Department of Exact Sciences, State University of Feira de Santana, Feira de Santana, BA, Brazil

[7]Institute of Mathematics and Statistics, Federal University of Bahia, Salvador, BA, Brazil

[8]Faculty of Epidemiology and Population Health, London School of Hygiene and Tropical Medicine, London, UK

**Acknowledgements** The authors thank the members of Rede CoVida's Epidemiology & Information Group for the work of identifying and collecting data related to COVID-19.

**Contributors** NJS, RCRS and RLF designed the study and analysis strategy. NJS, CASTS and MYTI obtained, documented and described the data. AJFF, CSST, ASR, FJOA and IRF carried out the literature search. NJS and EJP performed the data analysis. NJS, RCRS, AJFF, CSST, ASR, FJOA, IRF, ESP and MLB contributed to data interpretation. NJS, AJFF, CSST, ASR, FJO and IRF drafted the manuscript. RCRS, ESP, MYTI and MLB critically revised the manuscript. All authors read and approved the final manuscript.

**Funding** All authors are affiliated to the Centre for Data and Knowledge Integration for Health (CIDACS) that is funded and supported by MCTI/ NPq/MS/SCTIE/Decit/ Bill & Melinda Gates Foundation's GCE Brazil (OPP1142172), Wellcome Trust (202912/Z/16/Z), the Brazilian Health Surveillance Secretariat, Ministry of Health, Bahia State, Research Support Foundation of the State of Bahia (FAPESB), the Research and Project Funding Agency (FINEP), and the Secretariat of Science and Technology of the State of Bahia (SECTI). ESP is a fellow supported by the Wellcome Trust (213589/Z/18/Z).

**Competing interests** None declared.

**Patient consent for publication** Not required.

**Ethics approval** The study was conducted according to the guidelines laid down in the Declaration of Helsinki. As the study exclusively used publicly available de-identified data, ethics approval by a research ethics committee and informed consent are waived per Resolution n. 466/2012 of the National Health Council of Brazil's Commission of Ethics in Research.

**Provenance and peer review** Not commissioned; externally peer reviewed.

**Data availability statement** Data are available in a public, open access repository. Data are available upon reasonable request. Data are freely available without restriction at https://opendatasus.saude.gov.br/dataset/bd-srag-2020. Code book and analytical code will be made available upon request from the corresponding author.

**ORCID iDs**
Natanael de Jesus Silva http://orcid.org/0000-0003-3002-1032
Enny S Paixão http://orcid.org/0000-0002-4797-908X

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
