## [Reviewer comments · BMJ Open]

ARTICLE DETAILS

TITLE (PROVISIONAL)	Combined association of obesity and other cardiometabolic diseases with severe COVID-19 outcomes: a nationwide cross-sectional study of 21,773 Brazilian adult and elderly inpatients
AUTHORS	Silva, Natanael; Silva, Rita de Cássia; Ferreira, Andréa; Teixeira, Camila; Rocha, Aline; Alves, Flávia Jôse; Falcão, Ila; Pinto, Elizabete; Santos, Carlos Antônio; Fiaccone, Rosemeire; Ichihara, Maria Yury; Paixão, Enny; Barreto, Mauricio

VERSION 1 – REVIEW

REVIEWER	V Mohan Madras Diabetes Research Foundation
REVIEW RETURNED	18-Mar-2021

GENERAL COMMENTS	This is an interesting study by Natanael Silva and colleagues from Brazil where they report on severe COVID -19 outcomes in Brazilian adult and elderly inpatients. The authors report that the combined association of obesity and other cardiometabolic diseases results inverse in COVID-19 outcomes. The authors have to be congratulated on a nicely done study on a large number of participants (n = 21,942) with positive RT-PCR for SARS-CoV - 2. The authors conclude that the combined association of obesity, diabetes and all cardiovascular diseases with severe COVID – 19 outcomes is stronger in adults than in elders. Their study also shows an independent relationship of obesity with severe outcomes including a dose response association between severity of obesity and death in adults. The strengths of this study are that it describes the independent as well as the combined relationship of obesity with COVID – 19 severity in Brazil. The findings are of further interest because Brazil, not only had a large number of COVID – 19 (SARS-CoV – 2) cases but also some mutants have emerged there. Undoubtedly, there are some limitations as only hospitalised cases of severe COVID – 19 were included. However, even given these limitations, their findings are of interest because few studies have reported a dose response curve between severity of obesity and death due to COVID – 19. While diabetes, cardiovascular disease and respiratory disease have been associated with COVID – 19 deaths, the contribution of obesity to severity of COVID – 19 and to COVID – 19 deaths has not been well documented. In that sense, the authors have done the good job of collating this data. I was slightly confused by the some of the data presented by the authors and would like to clarify them. 1. In Table 1, under 'OB - Obesity', the 'Death' in the 'Yes' category was 4.5% while in the 'No' category, it was 4.7%. If the authors
---

	conclude that obesity was independently associated with death, how can one reconcile this from this Table? In the elderly group also, the percentages were 0.6 and 0.7%. 2. PR can be defined in the footnotes. 3. In Table 2, again, I was not able to understand, why diabetes and cardiovascular disease were included as variables among all the other variables when the grouping is already done according to the obesity alone, obesity + diabetes and / or CVD. I found the presentation confusing and difficult to interpret. 4. Having done this study, are there any suggestions that the authors would like to make for treatment of SARS-CoV-2 infection in those who are obese and have either diabetes or CVD as a comorbidity ?. 5. Are there any lessons that the rest of the world can learn from Brazil based on this extensive analysis? This would be useful for readers.
--	---

REVIEWER	José Pablo Suárez Llanos Hospital Universitario Nuestra Señora de la Candelaria, Endocrinology and Nutrition
REVIEW RETURNED	18-Mar-2021

GENERAL COMMENTS	It is a very interesting study, with a large sample size and on a highly relevant topic at the present time. Different points are exposed that the authors should explain and /or improve:  • Describe which pathologies were included in cardiovascular diseases. Was hypertension or hypertensive heart disease included?. This has been a risk factor associated with a worse evolution of COVID- 19. • There is some estimation of how the anthropometric values of the patients were determined?. Could it be described what percentage of these values was obtained by direct determination, estimation or referral by the patient? These are data that would provide relevant information for the analysis of the results due to the very possible underestimation of BMI obtained. • Check the homogeneity of letters and styles in the bibliographic references.
--

REVIEWER	Romina Buffarini Universidade Federal de Pelotas, Social Medicine
REVIEW RETURNED	22-Mar-2021

GENERAL COMMENTS	The authors addressed a very relevant topic for these days. They have a big sample with information directly taken from the hospitals records from all Brazil´s states, which is a strength. However, I think the manuscript needs a revision, specially Results and Discussion sections. Regarding the statistical analyses, the authors present a logistic regression with prevalence ratio (PR) as a measure of association. The estimation of logistic regression is odds ratio (OR). I would like to know if the authors use a special program to run logistic regression with a PR as a result. If it were the case, this would be explained in detail.
---

	I think that it would be clearer for the readership to show OR (ODDS RATIO), otherwise, it would be possible to use a Poisson regression with robust variance, which is appropriate for binary outcomes and its estimate is PR. Results section needs a major revision as the interpretations of the associations between exposures and outcomes are not clear. The models that tested the degree of obesity, were further adjusted for DM and CVD or these exposures were included in the obesity groups? Conclusion of the manuscript should be elaborated based on the results.
--	--

REVIEWER	F Mzayek University of Memphis, Division of Epidemiology, Biostatistics and Environmental Health
REVIEW RETURNED	28-Mar-2021

GENERAL COMMENTS	Re: Combined association of obesity and other cardiometabolic diseases with severe COVID-19 outcomes in 21,773 Brazilian adult and elderly inpatients COVID-19 is an important public health and health problem and many aspects of its impact on the population health are still poorly understood. The study, therefore, addresses an important topic and is well-written. There are few important points remain to be addressed. 1- It is not clear why the authors did not perform the same stratified analysis to assess the association of BMI categories with outcomes in the “elders” group. The analytical approach should be consistent, or a justification provided. 2- To this reviewer, some of the covariates should be exclusion criteria, especially indicators of serious comorbidities, such as chronic pulmonary diseases and immunosuppression. Adjusting for these factors in the multivariable models as simple (yes/no) variables is not enough, especially is that it does capture important aspects of their effects, such as duration, severity, and interaction with other comorbidities. Excluding these cases may decrease the generalizability of the findings a little bit but will increase the internal validity. 3- Although one of the categories of the main independent variable was “DM and/or CV disease”, there was no discussion of this subgroup results. For example, when comparing this subgroup with “obesity + DM and/or CV disease” subgroup the numbers suggest that the addition of obesity has more impact on the risk of all the outcomes in the adults group (Table 3). It would be interesting to discuss this observation. 4- Smoking is an important confounder for this research question. If information is available on smoking it should be adjusted for. 5- A short description how adjusted PRs were estimated from logistic models would be a nice addition to the methods. 6- Some comparison on the main characteristics between the analytical and the excluded study samples would help the readers to get an idea about the potential of selection bias (maybe a short supplemental table).
---

VERSION 1 – AUTHOR RESPONSE

Reviewer: 1

Dr. V Mohan, Madras Diabetes Research Foundation

Comments to the Author:

This is an interesting study by Natanael Silva and colleagues from Brazil where they report on severe COVID-19 outcomes in Brazilian adult and elderly inpatients. The authors report that the combined association of obesity and other cardiometabolic diseases results inverse in COVID-19 outcomes. The authors have to be congratulated on a nicely done study on a large number of participants (n=21,942) with positive RT-PCR for SARS-CoV-2. The authors conclude that the combined association of obesity, diabetes and all cardiovascular diseases with severe COVID-19 outcomes is stronger in adults than in elders. Their study also shows an independent relationship of obesity with severe outcomes including a dose response association between severity of obesity and death.

The strengths of this study are that it describes the independent as well as the combined relationship of obesity with COVID-19 severity in Brazil. The findings are of further interest because Brazil, not only had a large number of COVID-19 (SARS-CoV-2) cases but also some mutants have emerged there. Undoubtedly, there are some limitations as only hospitalized cases of severe COVID-19 were included. However, even given these limitations, their findings are of interest because few studies have reported a dose response curve between severity of obesity and death due to COVID-19. While diabetes, cardiovascular disease and respiratory disease have been associated with COVID-19 deaths, the contribution of obesity to severity of COVID-19 and to COVID-19 deaths has not been well documented. In that sense, the authors have done the good job of collating this data. I was slightly confused by the some of the data presented by the authors and would like to clarify them.

Thank you for the compliment. We really appreciate your comments and careful reading of our paper.

In Table 1, under 'OB - Obesity', the 'Death' in the 'Yes' category was 4.5% while in the 'No' category, it was 4.7%. If the authors conclude that obesity was independently associated with death, how can one reconcile this from this Table? In the elderly group also, the percentages were 0.6 and 0.7%.

The two-way relative frequencies presented in table 1 are crude estimates that do not account for the influence of other variables that can be related with both exposure and outcome, i.e. confounding variables such as sex, age, and other comorbidities. Our conclusions were made based on adjusted PRs and their 95%CI, which are more robust estimates for considering such confounding variables. In table 1, we observed an 33% increase in the prevalence of death in the adult group with obesity alone (PR 1.33, 95%CI 1.05-1.69) when compared with the group without obesity, DM, and CVD (none). In table 4, we also found an independent dose-response association between degrees of obesity and death in adults.

2. PR can be defined in the footnotes.

We included a definition for PR and 95%CI in the footnotes of the tables 2 and 3.

In Table 2, again, I was not able to understand, why diabetes and cardiovascular disease were included as variables among all the other variables when the grouping is already done according to the obesity alone, obesity + diabetes and/or CVD. I found the presentation confusing and difficult to interpret.

We removed diabetes and CVD among the other variables in tables 1 and 2.

Having done this study, are there any suggestions that the authors would like to make for treatment of SARS-CoV-2 infection in those who are obese and have either diabetes or CVD as a comorbidity? Our findings suggest important implications for the clinical care of patients with obesity and severe COVID-19, such as the increased need of critical care and invasive mechanical ventilation and higher risk of death among these patients. Also, the results support the inclusion of people with obesity, independently of age and other preexisting comorbidities, in the high-risk and vaccine priority groups for protection from SARS-CoV-2 infection. We made these suggestions in the conclusions (309-313).

Are there any lessons that the rest of the world can learn from Brazil based on this extensive analysis? This would be useful for readers.

The availability and use of data from our national influenza surveillance system (SIVEP-Gripe), which was created in 2005 driven by the H5N1 pandemic, improved over time and recently quick adapted to document COVID-19 cases, might be the biggest lesson and strength of our study. Severe acute respiratory syndrome is a condition of compulsory notification in Brazil, thus we had the opportunity to analyze a nationwide representative sample of patients hospitalized for severe COVID-

in both public and private hospitals. The large sample size and data availability allowed us to analyze the combined association of obesity, diabetes and cardiovascular disease with severe COVID-19 outcomes, separately by age groups and controlled by important confounding variables, e.g. underlying comorbidities. Therefore, the availability of health surveillance systems may be a lesson from Brazil that other countries can learn for obtaining routine and timely data to guide health systems and research in preparing and responding to pandemics before and during its course. These arguments are outlined in the strengths and limitations of the study (line 56-60, 279-282).

Reviewer: 2

Dr. José Pablo Suárez Llanos, Hospital Universitario Nuestra Señora de la Candelaria

Comments to the Author:

It is a very interesting study, with a large sample size and on a highly relevant topic at the present time. Different points are exposed that the authors should explain and /or improve:

Thank you very much for your comments and for taking time to review our paper.

Describe which pathologies were included in cardiovascular diseases. Was hypertension or hypertensive heart disease included? This has been a risk factor associated with a worse evolution of COVID- 19.

This information was obtained from a dichotomous question (yes/no) on the existence of 'any' chronic cardiovascular diseases, including hypertension, which were answered based on patient or family's report or medical diagnosis. We have made this point clearer in the methods (line 120-122).

There is some estimation of how the anthropometric values of the patients were determined? Could it be described what percentage of these values was obtained by direct determination, estimation or referral by the patient? These are data that would provide relevant information for the analysis of the results due to the very possible underestimation of BMI obtained.

We agree with you. However, there is no information in the dataset of how anthropometric measurements for each patient was obtained. This is discussed in the limitations (line 289-293).

Check the homogeneity of letters and styles in the bibliographic references.
We checked the references' letter fonts as suggested.

Reviewer: 3
Dr. Romina Buffarini, Universidade Federal de Pelotas

Comments to the Author:

The authors addressed a very relevant topic for these days. They have a big sample with information directly taken from the hospitals records from all Brazil's states, which is a strength. However, I think the manuscript needs a revision, specially Results and Discussion sections.

We are very grateful for your comments and for taking time to review our paper.

Regarding the statistical analyses, the authors present a logistic regression with prevalence ratio (PR) as a measure of association. The estimation of logistic regression is odds ratio (OR). I would like to know if the authors use a special program to run logistic regression with a PR as a result. If it were the case, this would be explained in detail. I think that it would be clearer for the readership to show OR (ODDS RATIO), otherwise, it would be possible to use a Poisson regression with robust variance, which is appropriate for binary outcomes and its estimate is PR.

Crude and adjusted estimates were interpreted based on the prevalence ratio (PR) and 95% confidence intervals (95%CI). These estimates were obtained from logistic models using delta method, function 'prLogisticDelta', which is implemented in R and available in the package 'prLogistic'. As suggested, we added the above explanation in the methods (line 161-162). OR would not be appropriate here because it could overestimate the associations due to the high prevalence of the analyzed outcomes.

Results section needs a major revision as the interpretations of the associations between exposures and outcomes are not clear.
We have revised the results sections as suggested.

The models that tested the degree of obesity, were further adjusted for DM and CVD or these exposures were included in the obesity groups?

The models that tested the degrees of obesity were adjusted for DM and CVD. We included a sentence in the methods to make that clearer (line 165-166). The table's footnotes also include a description of the variables which the models were adjusted for (Table 4).

Conclusion of the manuscript should be elaborated based on the results.

We have rewritten the conclusion of the study in both abstract and main text. We opportunely use this space to also make recommendations based on our results and related literature, given the high relevance of the topic today, to the clinical care of patients with obesity and severe COVID-19 and to the inclusion of individuals with obesity, independently of age and other preexisting comorbidities, in the high-risk and vaccine priority groups for protection from SARS-CoV-2 infection.

Reviewer: 4
Dr. F Mzayek, University of Memphis

COVID-19 is an important public health and health problem and many aspects of its impact on the population health are still poorly understood. The study, therefore, addresses an important topic and is well-written. There are few important points remain to be addressed.

Thank you very much for taking time to read our paper and for your thoughtful remarks.

1- It is not clear why the authors did not perform the same stratified analysis to assess the association of BMI categories with outcomes in the “elders” group. The analytical approach should be consistent, or a justification provided.

There are no BMI cutoff points for degrees of obesity in elders. The WHO cutoff points, which classify the degrees of obesity, are specific for adults. Thus, their use would not be appropriate for elders. As suggested, we included an explanation in the methods section (line 138-139).

2- To this reviewer, some of the covariates should be exclusion criteria, especially indicators of serious comorbidities, such as chronic pulmonary diseases and immunosuppression. Adjusting for these factors in the multivariable models as simple (yes/no) variables is not enough, especially is that it does capture important aspects of their effects, such as duration, severity, and interaction with other comorbidities. Excluding these cases may decrease the generalizability of the findings a little bit but will increase the internal validity.

We ran all analyses excluding the cases with chronic pulmonary diseases and immunosuppression. At the end of this letter, you can find the software outputs of the multivariate analyses for outcome ‘death’, firstly adjusting and secondly excluding for these two comorbidities. Comparing the results of the two approaches, we observed a little difference in the magnitude but not in the direction and significance of the associations, keeping the results the same. Valuing the external validity of the study, we have decided to keep the analyses adjusted for these comorbidities.

3- Although one of the categories of the main independent variable was “DM and/or CV disease”, there was no discussion of this subgroup results. For example, when comparing this subgroup with “obesity + DM and/or CV disease” subgroup the numbers suggest that the addition of obesity has more impact on the risk of all the outcomes in the adults group (Table 3). It would be interesting to discuss this observation.

We strongly agree with you. In both age groups, obesity alone and obesity combined with DM and/or CVD had more impact on the risk of all severe COVID-19 outcomes than the subgroup with DM and/or CVD. As suggested, we highlighted this finding in the results (lines 196-197, 202-203), discussion (lines 219-220), and also in the conclusions (line 306-307).

4- Smoking is an important confounder for this research question. If information is available on smoking it should be adjusted for.

Information on smoking is not available on Brazil’s influenza surveillance system (SIVEP-Gripe).

5- A short description how adjusted PRs were estimated from logistic models would be a nice addition to the methods.

We added the following description to the methods (line 161-162): “Crude and adjusted estimates were interpreted based on the prevalence ratio (PR) and 95% confidence intervals (95%CI). These estimates were obtained from logistic models using delta method, function ‘prLogisticDelta’, which is implemented in R and available in the package ‘prLogistic’.

6- Some comparison on the main characteristics between the analytical and the excluded study samples would help the readers to get an idea about the potential of selection bias (maybe a short supplemental table).

We have included a table on demographic and clinical characteristics of the overall study population and study samples included and excluded of the analysis in each outcome (Supplementary Table 1). Based on these characteristics, we observed that the analytical samples in each outcome were very similar to the overall study population and the excluded samples. We recall this table in the results (line 180-181) and in the discussion of the study strengths and limitations (line 295-299).

Combined association of obesity, DM and CVD with death in adults (20-59 years) - ADJUSTING for chronic pulmonary disease, immunosuppression, and other variables

```
. xi: logistic OBITOS_SRAG_COVID19 i.COMORB_OB_DCV_DM_2 i.SEXO IDADE_ANOS i.HEMATOLOGI_RECDE i.HEPATICA_RECDE i.ASMA_
> RECODE i.NEUROLOGIC_RECDE i.PNEUMOPATI_RECDE i.IMUNODEPRE_RECDE i.RENAL_RECDE if IDADE_ANOS<60
i.COMORB_OB_D~2  _ICOMORB_OB_0~3      (naturally coded; _ICOMORB_OB_0 omitted)
i.SEXO          _ISEX0_0~1           (naturally coded; _ISEX0_0 omitted)
i.HEMATOLOGI_~E _IHEMATOLOG_0~1      (naturally coded; _IHEMATOLOG_0 omitted)
i.HEPATICA_RE~E _IHEPATICA_0~1      (naturally coded; _IHEPATICA_0 omitted)
i.ASMA_RECDE    _IASMA_RECO_0~1      (naturally coded; _IASMA_RECO_0 omitted)
i.NEUROLOGIC_~E _INEUROLOGI_0~1      (naturally coded; _INEUROLOGI_0 omitted)
i.PNEUMOPATI_~E _IPNEUMOPAT_0~1      (naturally coded; _IPNEUMOPAT_0 omitted)
i.IMUNODEPRE_~E _IIMUNODEPR_0~1      (naturally coded; _IIMUNODEPR_0 omitted)
i.RENAL_RECDE   _IRENAL_REC_0~1      (naturally coded; _IRENAL_REC_0 omitted)

Logistic regression                               Number of obs =      6565
                                                  LR chi2(12)      =    306.62
                                                  Prob > chi2      =    0.0000
Log likelihood = -3914.9597                       Pseudo R2       =    0.0377
```

OBITOS_SRAG_COVID19	Odds Ratio	Std. Err.	z	P> z	[95% Conf. Interval]	
_ICOMORB_OB_1	1.371393	.1877468	2.31	0.021	1.04865	1.793467
_ICOMORB_OB_2	1.931055	.2383075	5.33	0.000	1.516176	2.45946
_ICOMORB_OB_3	1.174254	.0754805	2.50	0.012	1.035255	1.331916
_ISEX0_1	1.249078	.0700072	3.97	0.000	1.119134	1.39411
IDADE_ANOS	1.025611	.0033095	7.84	0.000	1.019145	1.032118
_IHEMATOLOG_1	1.298312	.2823011	1.20	0.230	.8478059	1.988208
_IHEPATICA_1	2.210089	.4003978	4.38	0.000	1.549527	3.152248
_IASMA_RECO_1	.6542568	.0798828	-3.47	0.001	.515014	.8311463
_INEUROLOGI_1	1.731473	.2591745	3.67	0.000	1.291232	2.321814
_IPNEUMOPAT_1	1.718178	.2283603	4.07	0.000	1.324148	2.229462
_IIMUNODEPR_1	1.511037	.1643156	3.80	0.000	1.220991	1.869984
_IRENAL_REC_1	2.468534	.2607494	8.55	0.000	2.006908	3.036344
_cons	.0913089	.0144804	-15.09	0.000	.0669151	.1245955

Combined association of obesity, DM and CVD with death in adults (20-59 years) - EXCLUDING pulmonary disease and immunosuppression cases

```
. xi: logistic OBITOS_SRAG_COVID19 i.COMORB_OB_DCV_DM_2 i.SEXO IDADE_ANOS i.HEMATOLOGI_RECODO i.HEPATICA_RECODO i.ASMA_
> RECODE i.NEUROLOGIC_RECODO i.RENAL_RECODO if IDADE_ANOS<60 & PNEUMOPATI_RECODO==0 & IMUNODEPRE_RECODO==0
i.COMORB_OB_D~2   _ICOMORB_OB_0-3   (naturally coded; _ICOMORB_OB_0 omitted)
i.SEXO           _ISEXO_0-1         (naturally coded; _ISEXO_0 omitted)
i.HEMATOLOGI_~E  _IHEMATOLOG_0-1   (naturally coded; _IHEMATOLOG_0 omitted)
i.HEPATICA_RE~E  _IHEPATICA_0-1   (naturally coded; _IHEPATICA_0 omitted)
i.ASMA_RECODO    _IASMA_RECODO_0-1 (naturally coded; _IASMA_RECODO_0 omitted)
i.NEUROLOGIC_~E _INEUROLOGI_0-1   (naturally coded; _INEUROLOGI_0 omitted)
i.RENAL_RECODO   _IRENAL_REC_0-1   (naturally coded; _IRENAL_REC_0 omitted)
```

```
Logistic regression           Number of obs   =      5889
                              LR chi2(10)         =      271.86
                              Prob > chi2         =      0.0000
Log likelihood = -3459.1607    Pseudo R2       =      0.0378
```

OBITOS_SRAG_COVID19	Odds Ratio	Std. Err.	z	P> z	[95% Conf. Interval]
_ICOMORB_OB_1	1.427713	.2083764	2.44	0.015	1.072524 1.900531
_ICOMORB_OB_2	1.99225	.2555487	5.37	0.000	1.549384 2.561702
_ICOMORB_OB_3	1.179184	.0817636	2.38	0.017	1.029343 1.350838
_ISEXO_1	1.227879	.0735616	3.43	0.001	1.091844 1.380864
IDADE_ANOS	1.02849	.0036134	8.00	0.000	1.021432 1.035596
_IHEMATOLOG_1	1.169052	.3038585	0.60	0.548	.7024098 1.945705
_IHEPATICA_1	2.486788	.5018294	4.51	0.000	1.674433 3.693259
_IASMA_RECODO_1	.6846794	.0879738	-2.95	0.003	.5322519 .8807596
_INEUROLOGI_1	1.781034	.2870229	3.58	0.000	1.298663 2.442574
_IRENAL_REC_1	3.001743	.3573534	9.23	0.000	2.377054 3.790599
_cons	.078872	.0136129	-14.72	0.000	.0562353 .1106206

Combined association of obesity, DM and CVD with death in elders (>=60 years)- ADJUSTING for chronic pulmonary disease, immunosuppression, and others variables

```
. xi: logistic OBITOS_SRAG_COVID19 i.COMORB_OB_DCV_DM_2 i.SEXO IDADE_ANOS i.HEMATOLOGI_RECODO i.HEPATICA_RECODO i.ASMA_
> RECODE i.NEUROLOGIC_RECODO i.PNEUMOPATI_RECODO i.IMUNODEPRE_RECODO i.RENAL_RECODO if IDADE_ANOS>=60
i.COMORB_OB_D~2   _ICOMORB_OB_0-3   (naturally coded; _ICOMORB_OB_0 omitted)
i.SEXO           _ISEXO_0-1         (naturally coded; _ISEXO_0 omitted)
i.HEMATOLOGI_~E  _IHEMATOLOG_0-1   (naturally coded; _IHEMATOLOG_0 omitted)
i.HEPATICA_RE~E  _IHEPATICA_0-1   (naturally coded; _IHEPATICA_0 omitted)
i.ASMA_RECODO    _IASMA_RECODO_0-1 (naturally coded; _IASMA_RECODO_0 omitted)
i.NEUROLOGIC_~E _INEUROLOGI_0-1   (naturally coded; _INEUROLOGI_0 omitted)
i.PNEUMOPATI_~E _IPNEUMOPAT_0-1   (naturally coded; _IPNEUMOPAT_0 omitted)
i.IMUNODEPRE_~E _IIMUNODEPR_0-1   (naturally coded; _IIMUNODEPR_0 omitted)
i.RENAL_RECODO   _IRENAL_REC_0-1   (naturally coded; _IRENAL_REC_0 omitted)
```

```
Logistic regression           Number of obs   =      9943
                              LR chi2(12)         =      630.04
                              Prob > chi2         =      0.0000
Log likelihood = -6239.6794    Pseudo R2       =      0.0481
```

OBITOS_SRAG_COVID19	Odds Ratio	Std. Err.	z	P> z	[95% Conf. Interval]
_ICOMORB_OB_1	1.70938	.4729793	1.94	0.053	.9938351 2.940107
_ICOMORB_OB_2	1.403671	.195708	2.43	0.015	1.068036 1.844781
_ICOMORB_OB_3	1.04718	.0550921	0.88	0.381	.944582 1.160922
_ISEXO_1	1.34805	.0585847	6.87	0.000	1.23798 1.467906
IDADE_ANOS	1.050947	.0026342	19.83	0.000	1.045797 1.056122
_IHEMATOLOG_1	1.195949	.2173925	0.98	0.325	.8375026 1.707808
_IHEPATICA_1	1.49568	.2778932	2.17	0.030	1.039172 2.15273
_IASMA_RECODO_1	.6567219	.074253	-3.72	0.000	.5261862 .8196407
_INEUROLOGI_1	1.491945	.1303667	4.58	0.000	1.257113 1.770643
_IPNEUMOPAT_1	1.33861	.1104586	3.53	0.000	1.138715 1.573595
_IIMUNODEPR_1	1.399895	.1464874	3.21	0.001	1.140313 1.718567
_IRENAL_REC_1	1.760196	.1495119	6.66	0.000	1.490251 2.079038
_cons	.0325451	.0062633	-17.80	0.000	.0223189 .0474567

Combined association of obesity, DM and CVD with death in elders (>=60 years) - EXCLUDING pulmonary disease and immunosuppression cases

```
. xi: logistic OBITOS_SRAG_COVID19 i.COMORB_OB_DCV_DM_2 i.SEXO IDADE_ANOS i.HEMATOLOGI_REC0DE i.HEPATICA_REC0DE i.ASMA_REC0DE i.NEUROLOGIC_REC0DE i.RENAL_REC0DE if IDADE_ANOS>=60 & PNEUMOPATI_REC0DE==0 & IMUNODEPRE_REC0DE==0
i.COMORB_OB_D~2 _ICOMORB_OB_0-3 (naturally coded; _ICOMORB_OB_0 omitted)
i.SEXO _ISEX0_0-1 (naturally coded; _ISEX0_0 omitted)
i.HEMATOLOGI_~E _IHEMATOLOG_0-1 (naturally coded; _IHEMATOLOG_0 omitted)
i.HEPATICA_RE~E _IHEPATICA__0-1 (naturally coded; _IHEPATICA__0 omitted)
i.ASMA_REC0DE _IASMA_REC0_0-1 (naturally coded; _IASMA_REC0_0 omitted)
i.NEUROLOGIC_~E _INEUROLOGI_0-1 (naturally coded; _INEUROLOGI_0 omitted)
i.RENAL_REC0DE _IRENAL_REC_0-1 (naturally coded; _IRENAL_REC_0 omitted)

Logistic regression Number of obs = 8676
LR chi2(10) = 570.73
Prob > chi2 = 0.0000
Log likelihood = -5478.0663 Pseudo R2 = 0.0495
```

OBITOS_SRAG_COVID19	Odds Ratio	Std. Err.	z	P> z	[95% Conf. Interval]	
_ICOMORB_OB_1	1.671931	.4815145	1.78	0.074	.9507651	2.94011
_ICOMORB_OB_2	1.442556	.2149615	2.46	0.014	1.077188	1.931852
_ICOMORB_OB_3	1.050838	.0606937	0.86	0.391	.938367	1.17679
_ISEX0_1	1.378917	.0639246	6.93	0.000	1.25915	1.510075
IDADE_ANOS	1.053715	.0028275	19.50	0.000	1.048188	1.059272
_IHEMATOLOG_1	1.048016	.2298315	0.21	0.831	.6818647	1.610786
_IHEPATICA__1	1.889255	.4156436	2.89	0.004	1.227503	2.907758
_IASMA_REC0_1	.6090122	.0763147	-3.96	0.000	.4763902	.7785548
_INEUROLOGI_1	1.468365	.1380231	4.09	0.000	1.221302	1.765409
_IRENAL_REC_1	1.770883	.1634301	6.19	0.000	1.477864	2.121998
_cons	.0265613	.0054831	-17.58	0.000	.0177228	.0398074

Independent association of degrees of obesity with death in adults (20-59 years) - ADJUSTING for chronic pulmonary disease, immunosuppression, and other variables

```
. xi: logistic OBITOS_SRAG_COVID19 i.OBES_IMC_CAT_WHO_2 i.SEXO IDADE_ANOS i.DIABETES_REC0DE i.CARDIOPATI_REC0DE i.HEMATOLOGI_REC0DE i.HEPATICA_REC0DE i.ASMA_REC0DE i.NEUROLOGIC_REC0DE i.PNEUMOPATI_REC0DE i.IMUNODEPRE_REC0DE i.RENAL_REC0DE if IDADE_ANOS<60
i.OBES_IMC_CA~2 _IOBES_IMC_0-3 (naturally coded; _IOBES_IMC_0 omitted)
i.SEXO _ISEX0_0-1 (naturally coded; _ISEX0_0 omitted)
i.DIABETES_RE~E _IDIABETES_0-1 (naturally coded; _IDIABETES_0 omitted)
i.CARDIOPATI_~E _ICARDIOPAT_0-1 (naturally coded; _ICARDIOPAT_0 omitted)
i.HEMATOLOGI_~E _IHEMATOLOG_0-1 (naturally coded; _IHEMATOLOG_0 omitted)
i.HEPATICA_RE~E _IHEPATICA__0-1 (naturally coded; _IHEPATICA__0 omitted)
i.ASMA_REC0DE _IASMA_REC0_0-1 (naturally coded; _IASMA_REC0_0 omitted)
i.NEUROLOGIC_~E _INEUROLOGI_0-1 (naturally coded; _INEUROLOGI_0 omitted)
i.PNEUMOPATI_~E _IPNEUMOPAT_0-1 (naturally coded; _IPNEUMOPAT_0 omitted)
i.IMUNODEPRE_~E _IIMUNODEPR_0-1 (naturally coded; _IIMUNODEPR_0 omitted)
i.RENAL_REC0DE _IRENAL_REC_0-1 (naturally coded; _IRENAL_REC_0 omitted)

Logistic regression Number of obs = 6565
LR chi2(14) = 348.65
Prob > chi2 = 0.0000
Log likelihood = -3893.9415 Pseudo R2 = 0.0429
```

OBITOS_SRAG_COVID19	Odds Ratio	Std. Err.	z	P> z	[95% Conf. Interval]	
_IOBES_IMC_1	1.359867	.1792141	2.33	0.020	1.050312	1.760656
_IOBES_IMC_2	1.463267	.2366458	2.35	0.019	1.06577	2.009017
_IOBES_IMC_3	1.907283	.3089642	3.99	0.000	1.388442	2.62001
_ISEX0_1	1.254807	.070541	4.04	0.000	1.123894	1.400969
IDADE_ANOS	1.024285	.0033363	7.37	0.000	1.017767	1.030845
_IDIABETES_1	1.506813	.0878047	7.04	0.000	1.344183	1.68912
_ICARDIOPAT_1	1.005743	.0580521	0.10	0.921	.8981628	1.126208
_IHEMATOLOG_1	1.340633	.2921763	1.35	0.179	.8745807	2.055039
_IHEPATICA__1	2.208115	.4016302	4.36	0.000	1.54596	3.153881
_IASMA_REC0_1	.6646316	.080933	-3.35	0.001	.5235151	.843787
_INEUROLOGI_1	1.774485	.2659584	3.83	0.000	1.322803	2.380399
_IPNEUMOPAT_1	1.750494	.2329324	4.21	0.000	1.348634	2.272098
_IIMUNODEPR_1	1.556608	.1689738	4.08	0.000	1.258285	1.925658
_IRENAL_REC_1	2.427977	.2582372	8.34	0.000	1.971114	2.990733
_cons	.0914691	.0144938	-15.09	0.000	.0670498	.124782

Independent association of degrees of obesity with death in adults (20-59 years) - EXCLUDING pulmonary disease and immunosuppression cases

```
. xi: logistic OBITOS_SRAG_COVID19 i.OBES_IMC_CAT_WHO_2 i.SEXO IDADE_ANOS i.DIABETES_RECODE i.CARDIOPATI_RECODE i.HEMAT
> OLOGI_RECODE i.HEPATICA_RECODE i.ASMA_RECODE i.NEUROLOGIC_RECODE i.RENAL_RECODE if IDADE_ANOS<60 & PNEUMOPATI_RECODE=
> =0 & IMMUNODEPRE_RECODE==0
i.OBES_IMC_CA~2  _IOBES_IMC_0-3      (naturally coded; _IOBES_IMC_0 omitted)
i.SEXO            _ISEXO_0-1          (naturally coded; _ISEXO_0 omitted)
i.DIABETES_RE~E  _IDIABETES_0-1     (naturally coded; _IDIABETES_0 omitted)
i.CARDIOPATI_~E  _ICARDIOPAT_0-1     (naturally coded; _ICARDIOPAT_0 omitted)
i.HEMATOLOGI_~E  _IHEMATOLOG_0-1     (naturally coded; _IHEMATOLOG_0 omitted)
i.HEPATICA_RE~E  _IHEPATICA_0-1     (naturally coded; _IHEPATICA_0 omitted)
i.ASMA_RECODE    _IASMA_RECO_0-1     (naturally coded; _IASMA_RECO_0 omitted)
i.NEUROLOGIC_~E  _INEUROLOGI_0-1    (naturally coded; _INEUROLOGI_0 omitted)
i.RENAL_RECODE   _IRENAL_REC_0-1     (naturally coded; _IRENAL_REC_0 omitted)
```

```
Logistic regression          Number of obs   =   5889
                             LR chi2(12)          =   317.66
                             Prob > chi2         =   0.0000
Log likelihood = -3436.2613   Pseudo R2      =   0.0442
```

OBITOS_SRAG_COVID19	Odds Ratio	Std. Err.	z	P> z	[95% Conf. Interval]
_IOBES_IMC_1	1.435745	.1954991	2.66	0.008	1.099444 1.874916
_IOBES_IMC_2	1.537301	.2593156	2.55	0.011	1.104527 2.139644
_IOBES_IMC_3	1.913327	.3268928	3.80	0.000	1.368867 2.674345
_ISEXO_1	1.234944	.0742283	3.51	0.000	1.097702 1.389345
IDADE_ANOS	1.027037	.003644	7.52	0.000	1.01992 1.034204
_IDIABETES_1	1.557109	.0947849	7.27	0.000	1.381988 1.75442
_ICARDIOPAT_1	.9861114	.0598626	-0.23	0.818	.875494 1.110705
_IHEMATOLOG_1	1.216423	.3173412	0.75	0.453	.7294954 2.028367
_IHEPATICA_1	2.493035	.5054964	4.51	0.000	1.675467 3.709548
_IASMA_RECO_1	.6965736	.089121	-2.83	0.005	.542079 .8950997
_INEUROLOGI_1	1.823957	.2943181	3.72	0.000	1.329421 2.502457
_IRENAL_REC_1	2.961369	.3551816	9.05	0.000	2.340997 3.746142
_cons	.0791735	.0136377	-14.72	0.000	.0564885 .1109685

VERSION 2 – REVIEW

REVIEWER	J Nolan Northern Kentucky University, Mathematics & Statistics
REVIEW RETURNED	25-May-2021

GENERAL COMMENTS	I reviewed both the manuscript as well as reviewer comments and responses from the previous round (particularly those related to the statistics). I believe the statistical methodology in use here seems sound. I believe the interpretations could be made better in the following way (and would require it): 1. In your adjusted analysis, the foci should be on the confidence intervals rather than the point estimates. In one particular glaring example you state that "Among elders, obesity without DM and CVD increased independently the prevalence of ICU admission by 40% (95%CI 1.07-1.82) and death by 67% (1.00-2.80)." For admission the correct statement would be to say that it is somewhere between 7% to 82% more likely (not that it is exactly 40% more likely, which is true for the sample, but who knows for the population). Likewise for death what you know is that it would be somewhere between equally likely (PR of 1) and 180% more likely. Here, you know nothing for certain since the value 1.0 is included in the PR. Based on the PR there could be a pretty big impact, or there
---

	could be no impact at all. ALL interpretations of PR should be changed to focus on the interval estimates, which have inferential value (rather than the sample point estimates which do not). Beyond this, I have only the following relatively minor suggestions: 2. One reviewer suggested the use of odds ratios rather than prevalence ratios. I believe you are correct to use prevalence here, however it would be appropriate to include a supporting reference (beyond the software reference) for these methods in your statistical analysis section. The following seems like it might work: Bruce, Nigel; Pope, Daniel; Stanistreet, Debbi. Quantitative methods for health research : a practical interactive guide to epidemiology and statistics (Second ed.). Hoboken, NJ. p. 16. ISBN 978-1-118-66526-8. OCLC 992438133. 3. Another reviewer suggested adding exclusion criteria. I agree with your response to that item, noting that you conducted a sensitivity analysis which is appropriate, and the results suggest no issues. I do believe that a statement to that effect should appear in your manuscript, and also that it might be appropriate to include the actual sensitivity analysis in supplemental materials.
--	--

REVIEWER	John Cursio The University of Chicago, Public Health Sciences
REVIEW RETURNED	02-Jun-2021

GENERAL COMMENTS	Thank you for this well-written and important article. I have a few comments and recommendations: The authors use prevalence rates instead of odds ratios in their approach, a few sentences added to justify this decision would help the reader. Was any stepwise selection used in the logistic models? If so, a description of the starting variables, selection method, and p-value criteria would be useful. Can prevalence rates be compared between adults and elders in table 3? This would be a nice addition to the supplementary material. For instance, non-invasive mechanical ventilation prevalence rates between adults and elders with OB is (2.13/1.22). Is this ratio significant? Other comparisons may reveal some interesting patterns. A sensitivity analysis using the BMI categories <18.5, 18.5-24.9, 25-29.9 could be useful. Someone with a BMI of 19 is much different than someone with 28.0. It may be possible to see a stronger dose-response relationship. This is not required in this paper but add it here as a possible direction for future research,
---

VERSION 2 – AUTHOR RESPONSE

Reviewer: 5

Dr. J Nolan, Northern Kentucky University

Comments to the Author:

I reviewed both the manuscript as well as reviewer comments and responses from the previous round (particularly those related to the statistics). I believe the statistical methodology in use here seems sound. I believe the interpretations could be made better in the following way (and would require it):

Thank you for your time in reviewing our manuscript and your valuable suggestions.

1. In your adjusted analysis, the focus should be on the confidence intervals rather than the point estimates. In one particular glaring example you state that "Among elders, obesity without DM and CVD increased independently the prevalence of ICU admission by 40% (95%CI 1.07-1.82) and death by 67% (1.00-2.80)." For admission the correct statement would be to say that it is somewhere between 7% to 82% more likely (not that it is exactly 40% more likely, which is true for the sample, but who knows for the population). Likewise for death what you know is that it would be somewhere between equally likely (PR of 1) and 180% more likely. Here, you know nothing for certain since the value 1.0 is included in the PR. Based on the PR there could be a pretty big impact, or there could be no impact at all. ALL interpretations of PR should be changed to focus on the interval estimates, which have inferential value (rather than the sample point estimates which do not).

Thank you for your explanation. Although we agree with your argument, which is very correct, interpretation focused on the interval estimates is not very applied in epidemiological studies. In our field, results are usually interpreted based on points estimates (the most likely value) given the statistical significance of the confidence intervals. In other words, a result is described and statistically significant when the confidence interval does not contain the null hypothesis value.

We have rewritten the mentioned sentence and made an observation regarding the borderline significant association of obesity (without DM and CVD) with death in elders: "Among elders, obesity without DM and CVD was independently associated with a higher prevalence of ICU admission (PR 1.40, 95%CI 1.07-1.82) and death (PR 1.67, 95%CI 1.00-2.80). It is worth to note that the association with death was borderline statistically significant".

Beyond this, I have only the following relatively minor suggestions:

2. One reviewer suggested the use of odds ratios rather than prevalence ratios. I believe you are correct to use prevalence here, however it would be appropriate to include a supporting reference (beyond the software reference) for these methods in your statistical analysis section. The following seems like it might work: Bruce, Nigel; Pope, Daniel; Stanistreet, Debbi. Quantitative methods for

health research: a practical interactive guide to epidemiology and statistics (Second ed.). Hoboken, NJ. p. 16. ISBN 978-1-118-66526-8. OCLC 992438133.

We added the reference as suggested.

3. Another reviewer suggested adding exclusion criteria. I agree with your response to that item, noting that you conducted a sensitivity analysis which is appropriate, and the results suggest no issues. I do believe that a statement to that effect should appear in your manuscript, and also that it might be appropriate to include the actual sensitivity analysis in supplemental materials.

We included this sensitivity analysis as supplementary material (Tables 2-3). We have also described this analysis in the methods (lines 199-202) and in results (lines 148-153).

Reviewer: 6

Dr. John Cursio, The University of Chicago

Comments to the Author:

Thank you for this well-written and important article. I have a few comments and recommendations:

We appreciate your time in reviewing our paper. Also, thank you for your thoughtful comments.

The authors use prevalence rates instead of odds ratios in their approach, a few sentences added to justify this decision would help the reader.

We included an explanation as suggested: "Given the high prevalence of the analyzed outcome that could overestimate odds ratio, crude and adjusted estimates were interpreted based on the prevalence ratio (PR) and 95% confidence intervals (95%CI)." A reference suggested by Reviewer 5 was also added: Bruce, Nigel; Pope, Daniel; Stanistreet, Debbi. Quantitative methods for health research: a practical interactive guide to epidemiology and statistics (Second ed.). Hoboken, NJ. p. 16. ISBN 978-1-118-66526-8. OCLC 992438133.

Was any stepwise selection used in the logistic models? If so, a description of the starting variables, selection method, and p-value criteria would be useful.

No stepwise selection was used. Confounding variables were theoretically decided based on the current state of knowledge on obesity and COVID-19 risk factors. We have just made that clearer in the manuscript and added some references: "Adjusted models included a set of confounding variables selected according to the current literature on obesity and severe COVID-9 risk factors".

VanderWeele TJ. Principles of confounder selection. Eur J Epidemiol. 2019 Mar;34(3):211-219.

doi: 10.1007/s10654-019-00494-6. Epub 2019 Mar 6. PMID: 30840181; PMCID: PMC6447501.

Popkin BM, Du S, Green WD, Beck MA, Algaith T, Herbst CH, Alsukait RF, Alluhidan M, Alazemi N, Shekar M. Individuals with obesity and COVID-19: A global perspective on the epidemiology and biological relationships. Obes Rev. 2020 Nov;21(11):e13128. doi:

10.1111/obr.13128. Epub 2020 Aug 26. PMID: 32845580; PMCID: PMC7461480.

Can prevalence rates be compared between adults and elders in table 3? This would be a nice addition to the supplementary material. For instance, non-invasive mechanical ventilation prevalence rates between adults and elders with OB is (2.13/1.22). Is this ratio significant? Other comparisons may reveal some interesting patterns.

Although this is a very interesting suggestion, we are unaware of any feasible measure that could handle a ratio of two prevalence ratios. That would definitely need further exploration!

A sensitivity analysis using the BMI categories <18.5, 18.5-24.9, 25-29.9 could be useful. Someone with a BMI of 19 is much different than someone with 28.0. It may be possible to see a

stronger dose-response relationship. This is not required in this paper but add it here as a possible direction for future research.

BMI in the SIVEP-Gripe is a mandatory information for patients diagnosed or self-reported with obesity only. Therefore, we do not have this variable complete for patients with BMI < 30. Instead, we used a dichotomous question (no/yes) on the existence of obesity to confirm the nutritional status of all patients. We made that clearer in the methods (lines 103-106) and discussed in the study limitations (lines 275-278).

VERSION 3 – REVIEW

REVIEWER	J Nolan Northern Kentucky University, Mathematics & Statistics
REVIEW RETURNED	30-Jun-2021
GENERAL COMMENTS	All requirements have been met in the paper, and my proposal would be to ultimately accept. However, I would point out to the

	authors that their response to my first point has two important flaws illustrating statistical misconception. First, to state that interpretations are usually "based on point estimates (the most likely value)" is poorly conceived. When using inferential statistics, there will be no "most likely value". Rather, all values within the confidence interval should be considered as equally plausible given the data (while values outside the CI are excluded). This is why interpretations should NOT be based on point estimates. The second component is the idea of describing a result as "statistically significant". The authors should keep in mind that statistical significance by itself never implies clinical relevance, and it is in fact the confidence interval that can get one to clinical relevance. In particular, for PR when you get a CI of 1.00 to 2.80, you do NOT have sufficient evidence to indicate clinical relevance, since it is entirely plausible that the true PR is 1.00. Your statement that "It is worth noting that the association with death was borderline statistically significant" should be changed to "It is worth noting that as 1.00 remains a plausible value for the PR, we cannot conclude that the association to death is clinically relevant (a larger sample would be needed to more fully address this)." Please fix that, as otherwise you are to some extent promoting the misconception that statistical significance automatically implies clinical relevance.
--	---

REVIEWER	John Cursio The University of Chicago, Public Health Sciences
REVIEW RETURNED	06-Jul-2021

GENERAL COMMENTS	Thank you for your appropriately updated manuscript. I do have a minor issue that can be addressed in both the methods and results. I was interested in why smoking was not included in the model? Is due to a lack of reporting or other data issues? Smoking and its relationship to obesity and other cardiovascular diseases is important in the research area presented in the manuscript. A description of why it was not considered would be helpful.
---

VERSION 3 – AUTHOR RESPONSE

Reviewer: 5
Dr. J Nolan, Northern Kentucky University

Comments to the Author:

All requirements have been met in the paper, and my proposal would be to ultimately accept. However, I would point out to the authors that their response to my first point has two important flaws illustrating statistical misconception. First, to state that interpretations are usually "based on point estimates (the most likely value)" is poorly conceived. When using inferential statistics, there will be no "most likely value". Rather, all values within the confidence interval should be considered as equally plausible given the data (while values outside the CI are excluded). This is why interpretations should NOT be based on point estimates. The second component is the idea of describing a result as "statistically significant". The authors should keep in mind that statistical significance by itself never implies clinical relevance, and it is in fact the confidence interval that can get one to clinical relevance. In particular, for PR when you get a CI of 1.00 to 2.80, you do NOT have sufficient evidence to indicate clinical relevance, since it is entirely plausible that the true PR is 1.00. Your statement that "It is worth noting that the association with death was borderline statistically significant" should be changed to "It is worth noting that as 1.00 remains a plausible value for the PR, we cannot conclude

that the association to death is clinically relevant (a larger sample would be needed to more fully address this)." Please fix that, as otherwise you are to some extent promoting the misconception that statistical significance automatically implies clinical relevance.

Thank you very much for your valuable feedback and explanations. As suggested, we have changed the statement indicated above and included it in the discussion section (line 224-226).

Reviewer: 6
Dr. John Cursio, The University of Chicago

Comments to the Author:

Thank you for your appropriately updated manuscript. I do have a minor issue that can be addressed in both the methods and results. I was interested in why smoking was not included in the model? Is due to a lack of reporting or other data issues? Smoking and its relationship to obesity and other cardiovascular diseases is important in the research area presented in the manuscript. A description of why it was not considered would be helpful.

Thank you very much for your suggestion and for taking time to review our work. We strongly agree with you about smoking. However, no information on health risk behaviors, such as smoking, is available on Brazil's influenza surveillance system (SIVEP-Gripe). We included in the manuscript a statement regarding the lack of smoking information in the SIVEP-Gripe dataset (line 283-291).

VERSION 4 – REVIEW

REVIEWER	J Nolan Northern Kentucky University, Mathematics & Statistics
REVIEW RETURNED	20-Jul-2021
GENERAL COMMENTS	No further concerns.
REVIEWER	John Cursio The University of Chicago, Public Health Sciences
REVIEW RETURNED	16-Jul-2021
GENERAL COMMENTS	The authors addressed my previous comments appropriately, and I have no further concerns. In my opinion, the manuscript is acceptable for publication.